# A new *Desmodesmus* sp. from the Tibetan Yamdrok Lake

**Jinhu Wang**[1☯]**, Qiangying Zhang**[1☯]**, Naijiang Chen**[2☯]**, Junyu Chen**[1]**, Jinna Zhou**[1]**, Jing Li**[1]**, Yanli Wei**[1]**, Duo Bu**[1]*

1 College of Science, Tibet University, Lhasa City, Tibet Autonomous Region, P. R. China, 2 Lianyungang Food and Drug Inspection and Testing Center, Lianyungang City, Jiangsu Province, P. R. China

☯ These authors contributed equally to this work.

* phudor@vip.163.com

**Data Availability Statement:** All relevant data are within the paper.

**Funding:** This research was funded by Research on Environmental Risk Management and Control of

## Abstract

Revegetation of exposed sub-soil, while a desirable recovery strategy, often fails due to extreme soil chemical properties, such as low organic matter and pH levels. Microalgae play a key role in maintaining water quality in the lakes and rivers on the Qinghai-Tibet plateau. Plateau microalgae have extensive application prospects in environmental purification, biotechnology, medicine and cosmetics, food industry, and renewable energy. To identify the high biomass of microalgae present in nature, microalgae with the greatest biomass were screened from natural water samples through filtration, pre-culture, and plate scribing separation. Following identification via 18S rRNA sequencing as for the *Desmodesmus* sp., we constructed a neighbor-joining phylogenetic tree. The novel *Desmodesmus* sp. from the Tibetan Yamdrok Lake were identified through polyphasic taxonomy. Simultaneously, the sequence of the experimental samples and the target species were shown different following the identification and analysis of SNP and InDel loci. The light-absorbing properties of plateau *Desmodesmus* sp. have been investigated previously. The characteristic absorption peak of *Desmodesmus* sp. on the plateau was measured at 689 nm in the visible spectrum using full wavelength scanning with a UV-Vis spectrophotometer. For *Desmodesmus* sp. which is prone to settling in the process of amplification culture. By monitoring the change trend of total nitrogen, total phosphorus, pH and electrical conductivity in algae solution system, we determined that the logarithmic growth phase and the best transfer window of *Desmodesmus* sp. were at 15–20 days. This study can provide basic research methods for the study of microalgae in high altitude areas, and lay a foundation for the later study and application of microalgae.

## Introduction

The Qinghai-Tibet Plateau is the highest plateau worldwide, with an average altitude of over 4000 m. It is characterized by a unique physical and geographical environment with cold temperatures, hypoxia, little precipitation, long sunshine, and strong solar radiation [1]. Rivers

Industrial Solid Waste Recycling Process in Low Temperature, Low Pressure and Anoxic Environment, grant number 2019YFC190410304, which was funded by Sub project of major R&D plan of the Ministry of Science and Technology; the central government supports the phased achievement funding of local university projects (ZCKJZ [2022] No. 1, [2021] No.1, [2020] No. 1 and [2019] No. 44); the funders had no role in study design, data collection and analysis, decision to publish, or preparation of the manuscript.

**Competing interests:** The authors have declared that no competing interests exist.

and lakes are distributed at different elevation gradients, and microalgae on the plateau play a key role in maintaining the water quality of these lakes and rivers [2].

Recently, wastewater treatment based on microalgae has attracted increasing attention due to its environmental friendliness and potential economic benefits [3]. Studies have shown that microalgae can remove various pollutant from wastewater, such as oxygen consuming pollutants, nitrogen, phosphorus, heavy metals, organic matter, and absorb harmful gases such as $NO_X$, $SO_X$, and $H_2S$ at a certain concentration. The biomass harvested from wastewater treatment can also be utilized as raw materials for biofuel, feed, and chemical compounds [4–9]. Microalgae can use $CO_2$ during photosynthesis as the only carbon source for heterotrophic growth and can use external carbon sources for heterotrophic growth to improve the biomass and oil content of microalgae [10, 11], which can greatly reduce global emissions of greenhouse gases such as $CO_2$ [12].

*Desmodesmus* sp. was collected from Yamdrok Lake, Langkazi County, Shannan City, Tibet Autonomous Region, named YH-1 and subsequently purified, amplified, and cultured. This area belongs to the semi-arid climate area of the plateau sub-cold zone, with lower annual average temperature, lower air density, lower oxygen content and strong solar radiation. The growth rate and cell components of vegetation and microalgae growing in this area are vastly different from those of crops in plain regions [13].

A microalgae from Yamdrok Lake with the highest biomass was screened through filtration, pre-culture, and the plate separation method. It was identified as *Desmodesmus* sp. using 18S rRNA sequencing. Genotyping was determined through PCR amplification and locus mutation (based on GATK analysis). The characteristic absorption peak was determined by studying the light absorption characteristics of *Desmodesmus* sp. on plateau. The linear relationship between the concentration of *Desmodesmus* sp., the absorbance as well as the turbidity were established using the optical density method, and a regression equation was established. By analyzing total nitrogen (TN), total phosphorus (TP), chemical oxygen demand (COD), pH, and electrical conductivity (EC) in the solution system during the course of amplification culture of *Desmodesmus* sp., the logarithmic growth end and the optimum transfer time of *Desmodesmus* sp. were determined.

Our results aid in understanding the isolation, purification, amplification, and culture methods of microalgae in high altitude areas. For some microalgae species that settle easily and affect the accuracy of absorbance value (such as *Desmodesmus* sp.) [14], the best transfer species or harvest time can be accurately determined by monitoring the TN, TP, COD, EC, and pH indicator values during the culture process. Using the linear relationship between microalgae concentration and absorbance as well as turbidity, the microalgae concentration can be determined, which can provide the basis for the follow-up large-scale production and application research. Using polyphasic taxonomy to identify *Desmodesmus* sp. from the Tibetan Yamdrok Lake. Through the identification and analysis of SNP and InDel sites to find the differences from that of the target species.

## Materials and methods

### Collection of microalgae

One litre of water was collected from the Yamdrok Lake (25 times by water collector), the phytoplankton were filtered through a phytoplankton net of 500 mesh, and the water sample was poured into a Teflon bucket of 50 L for preservation. Then, 200 g of sediment from the bottom of the lake was randomly collected at various points, and stored in a Teflon bottle with a breathable film tied at the mouth. Transport to the laboratory was completed within 6 h.

## Preparation method of culture medium

To prepare the liquid culture medium, we added 1.7 g BG 11 culture medium (Haibo Bio, Qingdao, China) into a 1000 mL conical flask, added distilled water to a constant volume, shook until the culture medium powder was completely dissolved. A 1 M HCl or NaOH solution was used to adjust the pH to 7.2. Then, we covered the bottle mouth of the conical flask with a rubber stopper, and then used a 0.22 μm sterilization membrane for filtration on a clean workbench, following which it was stored for later use.

To prepare the solid culture medium, we added 25 g of nutrient agar medium (Haibo Bio, Qingdao, China) to the liquid culture medium, heated it to 60˚C in water bath, and shook it until it was completely dissolved. Next, we used a 0.22 μm sterilization filter membrane for filtration on a clean workbench, following which it was stored it for later use. After the culture medium cooled to 35˚C, it was poured into a sterile petri dish and cooled continuously to obtain a solid culture medium.

## Pre-culture after filtration

Under open purification workbench conditions, the collected water samples were poured into the Brinell funnel and filtered under the suction filtration of the water ring vacuum pump using a 0.45 μm water-based filter membrane.

Each 100 L water sample was filtered thrice, with a new filter membrane being used each time. The replaced filter membrane was placed into a liquid culture medium prepared in advance for culture. Three groups of parallel cultures can be obtained using the three filtration steps. There was no need to filter the bottom mud collected from the water bucket; approximately 30 g of bottom mud was taken from each group and incubated into liquid culture medium separately, producing another three parallel groups. The top of the conical flask was sealed with a breathable sealing film to prevent external bacteria from entering.

The incubated samples were shaken thhrice a day in a light incubator with a temperature of 25 ± 1˚C (light-dark ratio of 12 h:12 h, light intensity of 2000 Lx), and each sample was randomly placed to receive light evenly. After culturing for 7–15 days, until the entire solution system was yellow-green or green, microscopic examination and dilution were carried out to separate microalgae strains [15].

## Isolation and purification of microalgae

When the purification workbench was turned on, the aforementioned fresh algal liquid at the end of logarithmic growth was dipped for approximately 10 days with the inoculation ring, and a dense line was drawn on the solid medium. Samples were inverted in a 25 ± 1˚C light incubator for culture (light dark ratio 12 h:12 h, light intensity 2000 Lx). The formation of microalgae strains with large biomass was observed after 7–10 days. Large algae colonies were collected for microscopic examination. Microscopic identification of algae indicated that the separation was successful [16, 17].

## The scale-up cultivation of *Desmodesmus* sp.

① In a 1,000 mL beaker, 500 mL BG 11 liquid medium was prepared according to the formula for BG 11 medium, and the pH was adjusted to 7.1. Next, 200 mL of BG 11 liquid medium was filled in each 500 mL conical flask, or 50 mL of BG 11 liquid medium filled in 150 mL conical flask.

② During step ①, the BG 11 culture medium should be filtered through a 0.22 μm sterilization filter membrane on a clean workbench, and then the bottle mouth should be bandaged for later use. Gas permeable membrane, sterile filter membrane, filter device, conical flask, and the inoculation ring should be sterilized by steam pressure sterilization pot at 101.33 kPa and 121˚C for 30 min before use.

③ On a clean workbench, the microalgae with good growth were selected using a sterile inoculation ring and incubated into 10 mL BG 11 liquid culture medium after sterilization and filtration at 25 ± 1˚C and a light intensity of 2,000 Lx (light ratio is 12 h:12 h). Next, the microalgae sample in medium was shook or stirred 2–3 times every 12 h and cultured for 7–15 days. When the biomass was obviously increased (>106/L), amplification culture can be carried out.

④ On a clean workbench, 10 mL of cultured algae liquid was added into the liquid medium containing 20 mL BG 11 for culture (1:2 amplification) at a temperature of 25 ± 1˚C and at 2,000 Lx (the light ratio is 12 h:12 h). Microalgae were cultured for 7–15 days by shaking or stirring 2–3 times every 12 h (bottle mouth was sealed with a gas permeable membrane and the biomass was obviously increased).

Normal culture conditions: temperature 25 ± 1˚C, light intensity 3,000 Lx (light ratio 12 h:12 h), shaking or stirring 2–3 times every 12 h. When the culture volume is greater than 1 L, consider adding stirring to assist the culture (culture time 7–15 days). When microalgae species with long culture period are used, the culture period can be appropriately prolonged.

⑥ Next, the culture solution from ⑤ was added to 150 mL BG 11 medium for amplification culture (1:5 amplification; culture period 15–20 days). When microalgae species with long culture period are used, the culture period can be appropriately extended to approximately 30 days.

⑦ For volumes of amplification culture medium greater than 1 L, the amplification culture conditions were 25 ± 1˚C, the light intensity was 3,000 Lx (light ratio is 12 h:12 h) and 100–200 r/min. When microalgae species with long culture period are used, the culture period can be appropriately extended to approximately 30 days.

⑧ After the logarithmic growth phase of green algae on the plateau, when the final optical density at 689 nm ($OD_{689}$) ≥ 0.70 (the maximum absorption wavelength of each algae species can be obtained through full-wavelength scanning using a UV-spectrophotometer), the algae solution after amplification was centrifuged or filtered and then washed with BG 11 culture solution for 3 times for later use [18, 19].

## Species identification of 18S rRNA

Using the extracted microalgae DNA as the template, primers 1143-510-2-F (`AATTGACG–GAAKGGCA`) and reverse 1637-510-2-R (`CGACGGGCGGTGTGTA`) were designed using the primer premier 5.0 software. The 18S rRNA gene was amplified using PCR. A total of 25 μL PCR reaction system was used, containing DNA template 1 μL, primers 1 μL each, dNTP 1 μL, Taq Buffer (with MgCl2) 2.5 μL, Taq enzyme 0.25 μL, and ddH2O to 25 μL. The PCR reaction conditions were: pre denaturation at 95˚C for 5 min, denaturation at 94˚C for 30 s, annealing at 63˚C for 30 s (0.5˚C per cycle), extension at 72˚C for 30 s, 10 cycles; Denaturation at 95˚C for 30 s, annealing at 58˚C for 30 s, extension at 72˚C for 30 s, 30 cycles; Recover and extend at 72˚C for 10 min and kept warm at 4˚C. After the PCR products were detected, the sequences

were blast compared using the GenBank database. Next, we constructed a neighbor-joining phylogenetic tree based on 18S rRNA sequences [16].

## Determination of amplification culture cycle

Microalgae consume nutrients such as N, P, and heavy metals during their growth process [20, 21]. Considering the easy sedimentation characteristics of *Desmodesmus* sp., the logarithmic phase of the growth, transfer, and harvest time of this specie were determined by detecting the change trends of TN, TP, pH, and EC in the culture solution of *Desmodesmus* sp. during the amplification culture stage; thus, preventing potential errors caused by the OD method [22].

The water quality multi parameter tester (KN—MUL20, Kenuokeyi instrument, Beijing) and intelligent digestion instrument (KN—HEA12, Kenuokeyi instrument, Beijing) were used to determine TN and TP. The water samples used to determine TN and TP were filtered through a 0.22 μm filter in advance. EC and pH were measured using a multi parameter test pen (PCSTestr—35, USA).

## Experiment on correlation between algae concentration and absorbance and turbidity

To determine algal concentration a blood cell counting plate and an optical microscope were used to count. The average value from three counts was computed and the average deviation was controlled within 10%. Before counting, the algal solution was vibrated on an oscillator to ensure the uniform dispersion of microalgae, and the initial algal concentration (PCS/L) was determined by taking the stock solution of *Desmodesmus* sp. culture solution as the test water sample.

Method for preparing water sample containing algae: Dilute 10 mL of the same volume of algal solution with ultrapure water successively in 20 mL, 40 mL, 80 mL, 100 mL, and 200 mL. Dilute it into a series of algal solutions with different concentrations at constant volume to allow cell counting and turbidity detection (there are three parallel samples). Dilute 10 mL of the same volume of algal solution with ultrapure water to 20 mL, 30 mL, 40 mL, 50 mL, and 100 mL respectively, and dilute it into a series of algal solutions with different concentrations at constant volume for cell counting and $OD_{689}$ measurements (there are three parallel samples).

According to the above values, fit the absorbance algae concentration curve and establish the linear regression equation. The same was done for algal turbidity and the concentration curve [23, 24].

## Morphology identification

**Methods for preparing samples for scanning electron microscopy.** Scanning electron microscopy (SEM) can be used to directly observe the three-dimensional structure of the specimen surface and reflect the morphological characteristics of various cell surfaces and fracture surfaces. The pre-treatment of SEM cell samples includes sampling, fixation, dehydration among others.

**Material preparation.** For bulk samples, the observation surface should be as flat as possible without affecting the purpose of the experiment and the size should be 2–3 mm thick and 5–6 mm long and wide.

**Fixing of samples.** Cultured cells were centrifuged at 800–1500 RPM for 8–15 min to enrich the precipitation. The precipitation was immersed in PBS (0.1M, without NaCI), the cells or tissues were rinsed several times, the supernatant was removed by centrifugation, and 2.5–3% glutaraldehyde precooled at 4˚C was added. Next, tissues were fixed at 4˚C for 4 h or

overnight (depending on the need of tissue size, this step can be extended), then the fixative was removed and samples were dipped into PBS (0.1M, without NaCI) 3–5 times, 15 min each time.

## Dehydration

Dehydration was performed with a gradient series of alcohol (30%, 50%, 70%, 80%, 90%, 95%, and 100%), 10–20 min at each time concentration (generally 15 min), then thoroughly dehydrate with 100% alcohol 1–2 times, and then incubated with isoamyl acetate (banana water) twice (20 min each time).

## Sample drying and conductive treatment

We used the critical drying method. Sample conduction treatment was applied using the vacuum spraying method (spraying should be uniform) after the completion of scanning electron microscope observation [25]. (Model: TEM Hitachi SU8010 (Japan), Leica DM500 biomicroscope (Germany))

## Preparation method of ultrathin section for transmission electron microscopy

**Materials and fixation.**   Cell sample: Cell mass should be at least half the size of a mung bean. The cells/bacteria were collected by centrifugation, the culture medium was discarded, 2.5% glutaraldehyde was added, and the cell mass was dispersed and stored at 4˚C. For adherent cells, the culture medium was first decanted, 2.5% glutaraldehyde was added, fixed at 4˚C for 15 min, scraped with cells, collected by centrifugation (fixative was retained), and stored at 4˚C. Samples were fixed for at least 4 h.

**Osmic acid fixation.**   Cells or tissues fixed with glutaraldehyde were rinsed trice with 0.1 M phosphate buffer (pH 7.2) for 15 min each time, and then fixed with 1% hic acid •0.1 M phosphate buffer (pH 7.2) for 2 h at room temperature (20˚C; the fixing time was adjusted appropriately for different samples). Next, samples were rinse trice with 0.1 M phosphate buffer (pH 7.2) for 15 min each time.

**Dehydration.**   The samples were dehydrated using 30%, 50%, 70%, 80%, 85%, 90%, 95%, and 100% (twice) alcohol, 15–20 min at each concentration (the dehydration time was appropriately extended for samples with more water content and thick cell membrane).

**Seepage.** The used osmotic agent was acetone: epoxy resin (2:1), acetone: epoxy resin (1:1), epoxy resin, 37˚C in the temperature box, 8–12 h each time.

**Embedding.**   The permeated samples were placed into capsules or embedded plates, the embedding agent epoxy resin was added, and the samples were polymerized at 60˚C for 48 h. The section thickness was 80–100 nm.

**Double staining.**   Uranium-lead double staining (2% uranyl acetate in saturated aqueous solution, lead citrate, stained for 15 min at room temperature) was performed. Sections were dried overnight at room temperature and imaged using electron microscopy [26, 27].

Model: TEM Hitachi 7800 (Japan)

Ultra-thin slicer: Leica (Germany) Model: EM UC7

**Identification and analysis of SNP and InDel loci.**   GATK software was used to identify base mismatches between transcriptome data and the reference genome of streptozoa (Assembly MH624152.1); thus identifying potential SNPs and InDel sites. GATK recognition criteria were as follows: (1) No more than three single base mismatches occurring within 35 bp; (2) The quality value after sequence depth standardization being greater than 2.0. Each sample was screened according to the above conditions and reliable SNP/InDel sites were finally

obtained [28]. SnpEff software was used to annotate the mutations according to the annotation information of the reference genome, and the distribution of the mutation sites on the genome structure was statistically analyzed [29]. According to the location of the mutation site on the reference genome and the gene location information on the reference genome, the region where the mutation occurred within the genome and the impact of the mutation (synonymous mutation or non-synonymous mutation) were predicted. SNP sites can be divided into Transition and Transversion according to the mode of base replacement [30].

## Results

### Species identification using SEM and STM

In our culture environment, the individual cells of alga were oval in shape and appeared as two-cell aggregates and single cells. The cell wall decoration of this alga was analyzed using SEM. The most prominent morphological features were uninterrupted rib-like decoration and warts without protrusions.

As shown in Fig 1A, the algal cells were arranged into two-cell aggregates. The two cells of the aggregate were of similar size, with a length and width in the range of 10.3–12.5 × 3.2– 3.4 μm, and they had a relatively high aspect ratio (3.5 ± 0.6 on average; Fig 1A). The lateral wall of the cell had a spinous protrusion that extended longitudinally as a whole and was approximately the same length as the cell (Fig 1A, arrow). The surface of the cell wall is similar to that of peanut shell and was densely ribbed in meridian and zonal directions. These ribs were formed by the thickening of the inner cell wall layer.

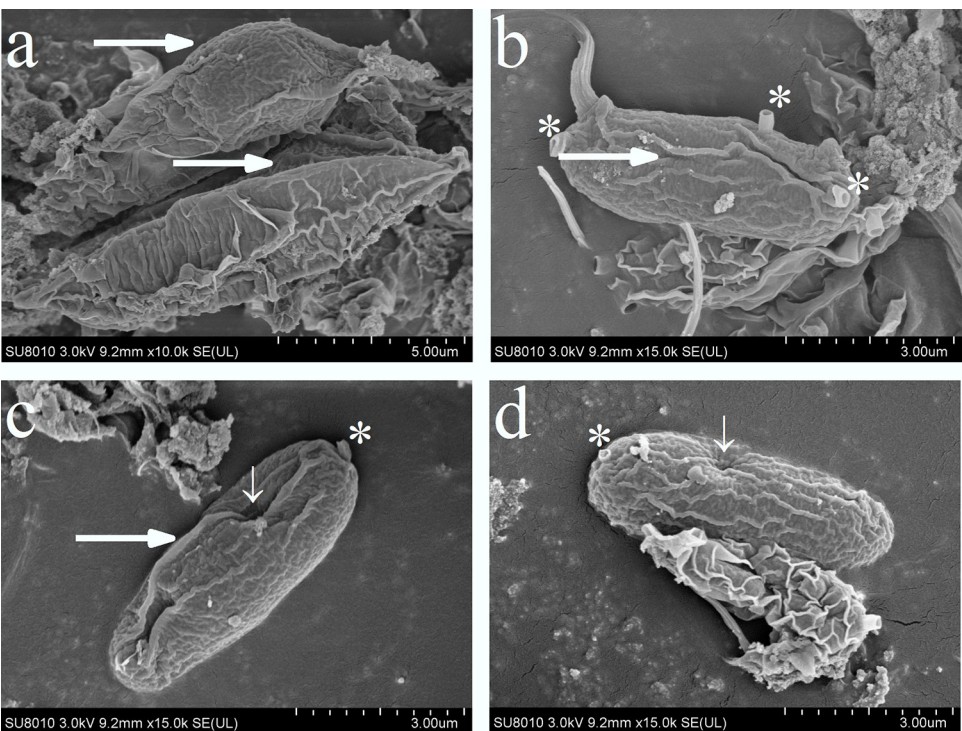

**Fig 1. Morphological graph based on SEM of *Desmodesmus* sp. (YH-1).** All cells studied retained chloroplasts, which contained thylakoids with an electron clear lumen (Fig 5C and 5D, arrows). In the chloroplast structure, the visible starch grains formed an incomplete sheath that was attached to the chloroplast envelope (Fig 5A and 5C). Simultaneously, a small number of plastid globules were observed in the microdomains of thylakoid blebs, which were structurally connected to the extracorporeal membrane of the thylakoid (Fig 5C and 5D).

The length and width of cells appearing as single cells ranged from 5.5–5.8 × 1.7–2.0 μm, with an average aspect ratio of 3.0 ± 0.2 (Fig 1B–1D). Among them, two types of single cells were observed, with thorns and without thorns. The cell wall also possessed a spinous process, extending longitudinally as a hole. The cell wall surface was mainly distributed with meridional rib-like ornamentation. Spiny single cells had chimney-like projections at the poles of the cell (Fig 1B, asterisk) and similar projections on the lateral wall of the cell. The outer wall of the projections appeared fairly smooth and vertical. The spiny single cell had a chimney-like protrusions only at one pole (Fig 1C and 1D, asterisk). There were clearly visible depressions on the cell wall of the spiny single cell, which are presumed to be gaps left by the thickening of the cell wall (Fig 1C and 1D, thin arrows).

## Species identification by 18S rRNA

Using the new generation sequencing method, DNA was extracted from the samples using the test kit produced by Sangon Biotech (Shanghai). The amplified PCR products were detected and purified through PCR amplification, agarose electrophoresis detection, gel recovery and other steps. The PCR products were sequenced using the 3730-xl sequencer produced by ABI (USA).

After PCR amplification, 18S rRNA was obtained. The length of the PCR amplification product: site sequencing primer was approximately 150–300 bp and 80–150 bp away from the site, while the exon detection primer was approximately 150 bp upstream and downstream of the exon. The PCR product band of target gene sequencing was generally shorter than 1200 bp. The login number used to analyze the sequence homology (NCBI) was PRJNA810921 and we discovered that our sequence was most closely related to *Desmodesmus* sp., with an 18S rRNA sequence homology of 99.6%. Through BLASt online comparison, we discovered 10 species sequences of *Desmodesmus* sp. species with a high similarity to YH-1 species (96.0%;).

To further analyze the evolutionary relationship of *Desmodesmus* sp., the 18S rRNA gene sequences of related species were downloaded from the GenBank database through sequence alignment and referring to NCBI annotation information. Simultaneously, according to the highly conserved characteristics of 18S rDNA, 32 sequences with high similarity and published sequences of algae genome related species were downloaded, the neighbor-joining phylogenetic tree was constructed using the Chloroella 18S rRNA (GenBank accession number: KC790435.1) and Chlorophyta 18S rRNA (GenBank accession number: HQ900842.1) outgroup sequences. The YH-1 and the *Desmodesmus* sp. (GenBank accession number: AB917136.1) clustered into one branch, and the two appear closely related. The *Desmodesmus* sp. with GenBank accession number KF673371.1 and MK541739.1 clustered into one branch. The *Desmodesmus* sp. with GenBank accession number: OK641939.1, MW077168.1, X73995.1, MK541740.1, MG022724.1, MW471025.1, MZ570911.1, AB917128.1, MW136451.1, MH624152.1, AB917097.1, and HM633188.1 clustered into one branch. After the aforementioned two branches clustered into one branch, they combined with the *Desmodesmus* sp. (GenBank accession number AB917136.1) and clustered into one branch, and then gather with YH-1 to form a large branch (Fig 2A and 2B).

## Identification and analysis of SNP and InDel loci

The Trimmomatic trimer was used to process the raw data obtained from sequencing in order to generate clean data. Data quality control statistics were as follows: Total Reads Count(#), 24615292; Total Bases Count (bp), 3597867603; Average Read Length (bp), 146.16; Q30 Bases Ratio (%), 92.96%; GC Bases Ratio (%), 60.57%. The most prominent types of variants in the entire sample were A-G, T-C, G-A, and C-T (Fig 3A).

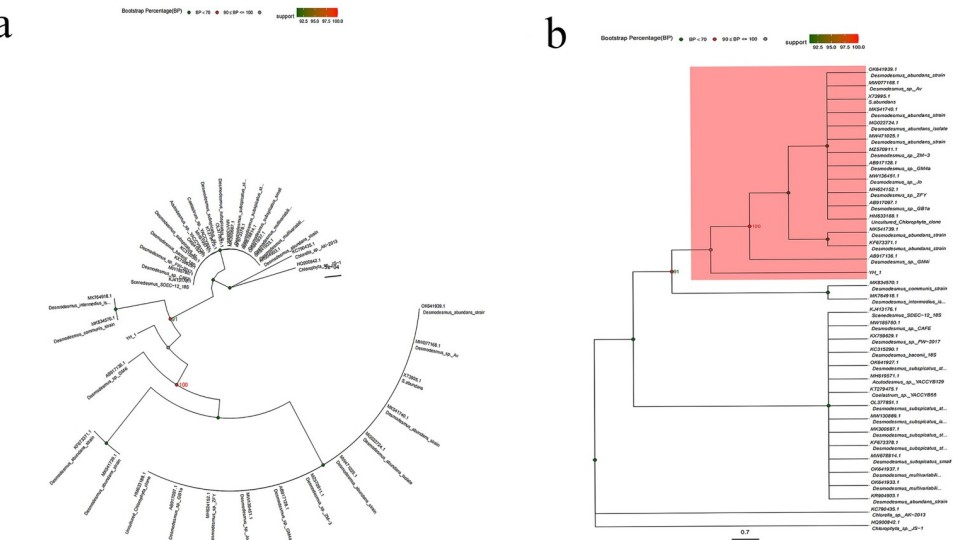

**Fig 2. Neighbor-joining phylogenetic tree based on 18S rRNA sequences.**

Considering 50 bp as the window for calculation, the distribution map of SNP/InDel in the entire gene mountain was constructed. The map indicates that there are great differences in the whole sequence range; thus, suggesting that the detected sample sequence is different from the target species sequence (Fig 3B).

The outer circle represents the size of the entire gene, with 50 bp as a small window. The second circle represents the distribution of each variant on the whole gene and its RMS mapping quality (MQ). The larger MQ represents the variant accuracy. The third circle represents the sequencing depth of each variant and the last circle represents the mutation conversion type. SNP mutation between purines or pyrimidines, are represented by 1, while SNP mutation between purines and pyrimidines, are represented by 0. SNP mutation between purines and pyrimidines, are represented by– 1; Where A > T and T > A are represented as the same type in the figure (Fig 3C).

## Determination of amplification culture cycle

The TP monitoring data indicates that the best transfer time for *Desmodesmus* sp. is 15–20 days after amplification. After 20 days, microalgae undergo apoptosis, which leads to increased TP index in the solution. The main reason for the increase in TP is the release of organic matter and metal elements caused by the decomposition of microalgae cells. Therefore, for the duration of 20 days, it is necessary to replant or harvest again (*Fig* 4A).

The TN monitoring data suggests that best transfer time for *Desmodesmus* sp. is 15–20 days after amplification. After 20 days, microalgae undergo apoptosis, which leads to increased TN index in the solution. The main reason for the increase in TN is the release of organic matter and metal elements caused by the decomposition of microalgae cells. Therefore, over 20 days, it is necessary to replant or harvest again (*Fig* 4B).

The pH value of microalgae fluctuates during the amplification period, mainly because the growth process of microalgae relies on $CO_2$ consumption from the atmosphere. The consumption of $CO_2$ in the process of dissolution and photosynthesis in water will induce the change in the pH value of water. We observed that between 15–23 days, microalgae reached a balance between $CO_2$ adsorption and consumption during the growth process; thus, reaching a stable

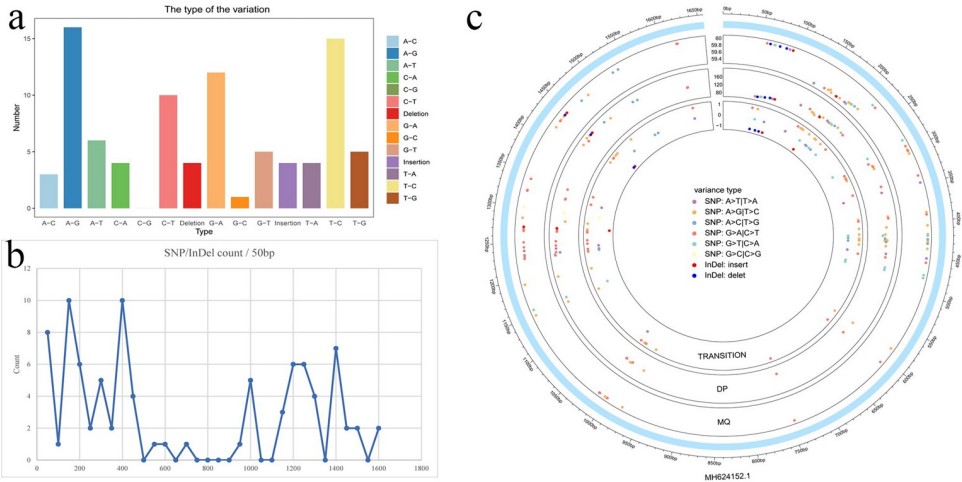

**Fig 3.** (a) Type and number of individual variants in the entire sample. (b) SNP/InDel distribution in the entire gene mountain was calculated using a 50 bp window. (c) Distribution of individual variants across the entire gene in the entire sample.

growth state. The pH monitoring data indicates that the best transfer time for *Desmodesmus* sp. is 15–20 days after amplification. After more than 20 days, it is necessary to replanted or harvested again (*Fig* 4C).

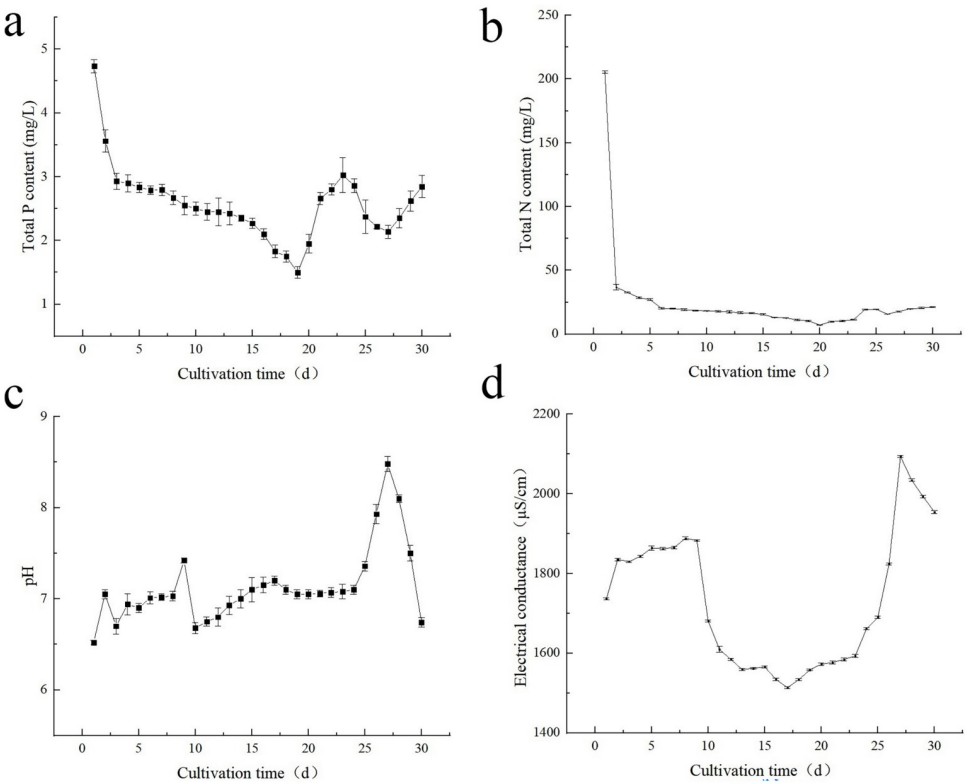

**Fig 4.** (a) Trend diagram of TP during the microalgae amplification stage. (b) Trend diagram of TN during the microalgae amplification stage. (c) Variation trend of pH value during the microalgae amplification and culture stage. (d) Variation trend of EC during the microalgae amplification stage.

The EC value of microalgae fluctuates during the amplification culture process, mainly because the growth process of microalgae relies on the consumption of metal nutrients from the culture medium; thus, resulting in a decrease in EC. Simultaneously, the natural growth of microalgae leads to an increase in EC, especially after the logarithmic growth phase as cell wall breaking will lead to the release of metal elements and ultimately an increase in EC. Based on the EC values obtained in this study, the best transfer time for *Desmodesmus* sp. is 15–20 days after amplification. After more than 20 days it is necessary to replanted or harvested again (*Fig* 4D).

### Study on light absorption characteristics

After full wavelength scanning using a UV-Vis spectrophotometer, the stock solution from the *Desmodesmus* sp. culture solution demonstrated no obvious absorption peak in the UV region, while an obvious absorption peak was observed at 689 nm, that is, its characteristic absorption peak. The supernatant (background liquid) obtained following centrifugation of microalgae liquid culture medium was analyzed similarly, and the absorption curve was relatively stable without an obvious absorption peak; therefore, the characteristic absorption wavelength was concluded to be 689 nm. After the characteristic absorption peak was obtained, the relationship between the concentration of *Desmodesmus* sp. and the absorbance of culture solution was investigated (S1 Fig).

### Relationship between Desmodesmus sp. concentration and absorbance of culture solution

Our results indicate a good linear relationship between the concentration of *Desmodesmus* sp. in the culture solution and the absorbance within a certain concentration range ($10^6$–$10^8$/L) at the maximum absorption wavelength of 689 nm. The concentration of microalgae in the culture process can be accurately derived based on absorbance values (S2 Fig).

### Relationship between Desmodesmus sp. concentration and turbidity of culture solution

Our results indicate a good linear relationship between the concentration of *Desmodesmus* sp. in the culture solution and the turbidity within a certain concentration range ($10^6$–$10^8$/L). The concentration of microalgae in the culture process can be accurately derived based on turbidity (S2 Fig).

## Discussion

Using water samples collected from Yamdrok Lake (4400 meters above the sea level), following filtration, pre-culture, as well as plate separation methods of which, we obtained microalgae strains with the maximum biomass after separation and purification. Following amplification and culture, the *Desmodesmus* sp. were microscopically observed and one microalgae strain with a relatively high biomass was selected.

By first filtrating and then pre-culturing before the plate separation steps, one can benefit from using simple equipment, simple operation, and reduced workload. The separated samples are likely to be obtained from two or more cells. The isolated single cell species of microalgae might not be the target species; however, other species or new species may be isolated. The used method is especially suitable for the preliminary separation of water samples from natural water areas, without being limited by the species and size of microalgae. Here we showed that we successfully isolated *Desmodesmus* sp. for the first time from Yamdrok Lake, Tibet.

## Using polyphasic taxonomy to identify *Desmodesmus* sp. from Tibetan Yamdrok Lake

The surface morphology of the cell wall and the presence of spines are important features that distinguish *Scenedesmus* sp. from *Desmodesmus* sp. Species belonging to the Scenedesmus family have no spines and have undecorated cell walls. However, two species types from the *Desmodesmus* family, namely spiny species and small spiny non-spiny species. The most prominent morphological features were uninterrupted rib-like decoration and warts without protrusions, which were different from the morphological features of other Streptomyces cells in the literature [31–35].

The cells investigated in this study retained their ultrastructure intact (Fig 5A) and the cellular ultrastructure shown in Fig 5 is typical of the observed microalgae [36–39]. By analyzing the sequence homology from NCBI, we observed that the isolated microalgae was closely related to *Desmodesmus* sp., with a 18S rRNA sequence homology of 99.6%. Following BLASt comparison, we discovered 10 sequences of *Desmodesmus* sp. with high similarities to the YH-1 species (96.0%), which form a separate branch on the neighbor-joining phylogenetic tree

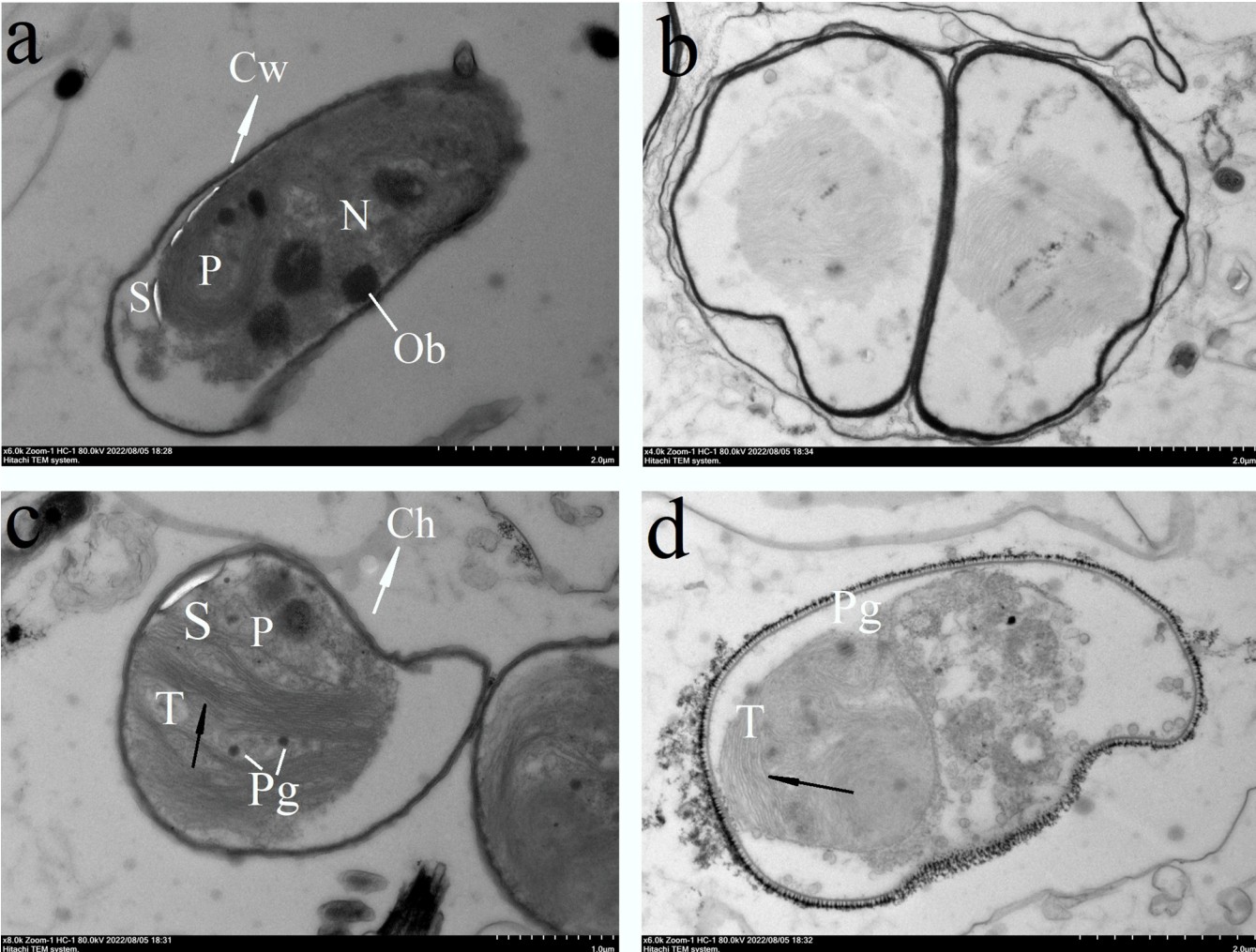

**Fig 5. Ultrastructure of *Desmodesmus* sp. cells.** (a) Overall view of the cell. (b–d) Regions of chloroplasts. Ch: Chloroplast; Cw: cell wall; N: nucleus; Ob: oil body; P: protein nucleus; Pg: plastid pellet; S: starch grain; T: cell nucleus.

[15]. ThroughBased on the polyphasic taxonomic identification method, we can conclude that the *Desmodesmus* sp. identified in Yamdrok Lake is a new species.

### The identification and analysis of SNP and InDel sites indicate that the sequence of the experimental samples is different from that of the target species

The distribution and mutation types of these SNPs/InDels are illustrated using graphs in Fig 3 and shows that our SNP/InDel sequencing was of high quality and that the sequence of the sample was different from that of the target sequence. Furthermore, the mutation quality of the experimental sample and the reliability were high. The mutation types of the experimental samples were distributed throughout the entire sequence, and there were numerous types of mutations with very high quality. The number of SNPs and InDels in the entire sequence, the intuitive number of mutations at each position, the high-quality evidence, and the refined mutation type show that our experimental samples are significantly different from the target sequence at genetic level, which indicates that our experimental samples represent a newly discovered species of *Desmodesmus* sp.

### Study on the growth cycle of amplification culture of *Desmodesmus* sp.

Next, we showed that the maximum characteristic absorption peak of *Desmodesmus* sp. was 689 nm. Furthermore, we showed that the culture medium of *Desmodesmus* sp. had a good linear relationship with the absorbance and turbidity.

During the culture process, Desmodesmus sp. exhibit obvious sedimentation; thus, being associated with errors when determining the logarithmic growth end through simple optical density method. By measuring the change trend of TN, TP, pH, and EC in the water during the amplification culture of Desmodesmus sp., we accurately assessed the end of the logarithmic growth phase as well as the best time for species transferring and picking [40].

After a 1:5 amplification culture, the changes in TN, TP, pH, and EC levels were monitored in real time for 30 days. We concluded that the best transfer time was 15–20 days after the amplification period. After more than 20 days, it is necessary to replanted or pick the microalgae again. Our findings provide novel insights into the methods used to expand the culture of microalgae.

### Conclusions

Novel species of *Desmodesmus* sp. from Yamdrok Lake, Tibet were identified using polyphasic taxonomy. Simultaneously, the S18 rRNA sequences of the experimental samples and the target species were shown to be different by identifying and analyzing the SNP and InDel loci.

By measuring the change trend of TN, TP, pH, and EC of *Desmodesmus* sp. during the scale-up cultivation, the end of logarithmic growth as well as the best time of seed transfer and harvest can be accurately assessed.

*Desmodesmus* sp. has its characteristic absorption peak at 689 nm. Within a certain concentration range, the algae concentration in the culture solution has a good linear relationship with absorbance and turbidity; therefore, absorbance or turbidity can be used to determine the algae concentration during culture.

During the experiment, the bubbles produced by the shaking process and the sedimentation of *Desmodesmus* sp. also have a certain influence. When measuring the absorbance and turbidity, it is necessary to shake the algal liquid evenly to reduce the detection error caused by

sedimentation of microalgae. Through the basic research of *Desmodesmus* sp., we provided the groundwork for the study of other microalgae.

## Supporting information

**S1 Table. Genotyping results.**
(XLSX)

**S1 Fig.** (a) Full wavelength scanning spectrogram of *Desmodesmus* sp. (b) Full wavelength scanning spectrum of background solution.
(TIF)

**S2 Fig.** (a) Relationship between microalgae concentration and absorbance of culture solution. (b) Relationship between microalgae concentration and turbidity of culture solution.
(TIF)

## Acknowledgments

We would like to thank Sangon Biotech (Shanghai) Co., Ltd., for the excellent 18S rRNA sequencing and analysis.

## Author Contributions

**Conceptualization:** Jinhu Wang, Qiangying Zhang, Naijiang Chen, Duo Bu.

**Data curation:** Jinhu Wang, Qiangying Zhang, Naijiang Chen, Junyu Chen, Jinna Zhou, Jing Li, Yanli Wei.

**Formal analysis:** Jinhu Wang, Qiangying Zhang, Naijiang Chen, Junyu Chen, Jinna Zhou, Jing Li, Yanli Wei.

**Funding acquisition:** Duo Bu.

**Investigation:** Duo Bu.

**Methodology:** Jinhu Wang, Qiangying Zhang, Naijiang Chen.

**Resources:** Jinhu Wang, Qiangying Zhang, Naijiang Chen, Junyu Chen, Jinna Zhou, Jing Li, Yanli Wei.

**Supervision:** Qiangying Zhang, Duo Bu.

**Writing – original draft:** Jinhu Wang, Qiangying Zhang, Naijiang Chen, Duo Bu.

**Writing – review & editing:** Jinhu Wang, Qiangying Zhang, Naijiang Chen, Junyu Chen, Jinna Zhou, Jing Li, Yanli Wei, Duo Bu.

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
