## [Decision Letter · Decision Letter 0]

12 Jul 2022

PONE-D-22-12548A new Desmodesmus sp. from Tibetan Yamdrok LakePLOS ONE

Dear Dr. Bu,

Thank you for submitting your manuscript to PLOS ONE. After careful consideration, we feel that it has merit but does not fully meet PLOS ONE’s publication criteria as it currently stands. Therefore, we invite you to submit a revised version of the manuscript that addresses the points raised during the review process.

We look forward to receiving your revised manuscript.

Kind regards,

Vandana Vinayak, PhD

Academic Editor

PLOS ONE

Journal Requirements:

3. PLOS requires an ORCID iD for the corresponding author in Editorial Manager on papers submitted after December 6th, 2016. Please ensure that you have an ORCID iD and that it is validated in Editorial Manager. To do this, go to ‘Update my Information’ (in the upper left-hand corner of the main menu), and click on the Fetch/Validate link next to the ORCID field. This will take you to the ORCID site and allow you to create a new iD or authenticate a pre-existing iD in Editorial Manager. Please see the following video for instructions on linking an ORCID iD to your Editorial Manager account: https://www.youtube.com/watch?v=_xcclfuvtxQ.

6. Thank you for stating the following financial disclosure:

“This research was funded by Research on environmental risk management and control of industrial solid waste recycling process in low temperature, low pressure and anoxic environment, grant number 2019YFC190410304 witch was funded by Sub project of major R&D plan of the Ministry of science and technology. This research was also funded by The central government supports the phased achievement funding of local university projects (ZCKJZ [2021] No. 1, [2020] No.1, [2019] No. 44 and [2018] No.54).”

Reviewers' comments:

Reviewer's Responses to Questions

**Comments to the Author**

1. Is the manuscript technically sound, and do the data support the conclusions?

Reviewer #1: Partly

Reviewer #2: Partly

2. Has the statistical analysis been performed appropriately and rigorously? 

Reviewer #1: Yes

Reviewer #2: No

3. Have the authors made all data underlying the findings in their manuscript fully available?

Reviewer #1: Yes

Reviewer #2: Yes

4. Is the manuscript presented in an intelligible fashion and written in standard English?

Reviewer #1: Yes

Reviewer #2: No

5. Review Comments to the Author

Reviewer #1: The manuscript titled "A new Desmodesmus sp. from Tibetan Yamdrok Lake" is interesting. However the following corrections are recommended:

1. How did you claim that the Desmodesmus sp. from Tibetan Yamdrok Lake is new? Validate the statement with concrete facts and findings.

2. Taxonomic keys are missing. Refer to the taxonomic credential with reference to the microalgae and the previous report.

3. The discussion needs to be strengthened in the light of the relevant literature available on the Desmodesmus sp.

4. The paper needs to corrected throughout to develop the flow and findings of the paper in the sequential pattern.

5. The paper requires major modification.

Reviewer #2: The authors describe their findings on a Desmodesmus species from Tibetan Yamdrok Lake in Tibet, China via a series of methods including strain isolation, molecular identification and determination of culture conditions. Totally speaking, the authors should clearly emphasize the novelty of this finding in the manuscript. Besides, the following comments are also for your references to improve the quality.

1. ‘sp.’ should not in italic form.

2. The resolution of figures is too low to provide information.

3. The data should be carried out in triplicate, and error bars should be provided.

4. In the M&M section, the general process of experimental methods should be concise by referring the published paper.

5. The initial value of TN in Figure 3c should be 200-260 ppm. Please check.

6. There is almost no discussion within the manuscript. The authors should emphasize why they perform this work. What is the significance?

7. The format of the references needs to be unified.

8. There are spelling, grammar and formatting errors in this article, and it should be applied to a professional editing service for language improvement.

6. PLOS authors have the option to publish the peer review history of their article (what does this mean?). If published, this will include your full peer review and any attached files.

Reviewer #1: **Yes: **Archana Tiwari

Reviewer #2: No

---

## [Author Response · Author response to Decision Letter 0]

28 Aug 2022

Dear Vandana Vinayak,

Thanks for providing us with this great opportunity to submit a revised version of our manuscript. We appreciate the detailed and constructive comments provided by the reviewers. We have carefully revised the manuscript by incorporating all the suggestions by the review panel.

We have read the reviewers’ and your comments carefully and have made revision which marked in red in the manuscript. We have tried our best to revise our manuscript according to the comments. Attached please find the revised version, which we would like to submit for your kind consideration. Here, we would like to explain the changes briefly as follows: We have rewritten the article based on your comments, placing more emphasis on the methods, results, discussion and conclusion. In the remainder of this letter, we discuss each of your comments individually along with our corresponding responses. We have written a point-by-point response letter for two reviewers, you can see the details at the end of this letter. In all, we found these comments are quite helpful. And special thanks to you and reviewers for your good comments again.

Reply to Reviewer #1

Dear Reviewer,

Thank you very much for giving us an opportunity to revise our manuscript, and we also appreciate you very much for your positive and constructive comments and suggestions on our manuscript.

Comments: “The manuscript titled "A new Desmodesmus sp. from Tibetan Yamdrok Lake" is interesting. However the following corrections are recommended:

1. How did you claim that the Desmodesmus sp. from Tibetan Yamdrok Lake is new? Validate the statement with concrete facts and findings.

2. Taxonomic keys are missing. Refer to the taxonomic credential with reference to the microalgae and the previous report.

3. The discussion needs to be strengthened in the light of the relevant literature available on the Desmodesmus sp.

4. The paper needs to corrected throughout to develop the flow and findings of the paper in the sequential pattern.

5. The paper requires major modification.”

Thank you very much for your opinion. We have rewritten the article based on your comments, placing more emphasis on the methods, results, discussion and conclusion. In the remainder of this letter, we discuss each of your comments individually along with our corresponding responses. To facilitate this discussion, we first retype your comments in italic font and then present our responses to the comments. 

Comment 1: How did you claim that the Desmodesmus sp. from Tibetan Yamdrok Lake is new? Validate the statement with concrete facts and findings.

Response 1: Thank you very much for your professional review, which was a great help in revising the article. The morphological identification of microalgae is the key and difficult point of the Desmodesmus sp.. We have highlighted this work in the article, and the method, results and discussion are presented in the article. Through the identification and analysis of SNP and InDel sites, it was further clarified that there were obvious SNP and InDel sites differences between theDesmodesmus sp. found in Yamdrok Lake and the existing algal strains of the same species, which proves its new characteristics in adapting to the special environment of high altitude., which from the side verified its new characteristics in adapting to the special environment of high altitude. Specifically, databases for identification were described in the “Result” and “Discussion”. 

Comment 2: Taxonomic keys are missing. Refer to the taxonomic credential with reference to the microalgae and the previous report.

Response 2: Your comment reminds us to focus on the description of taxonomic credential, which is crucial to the structure of the article. In this modification, we added the experiments of STM and SEM and described the morphology of Desmodesmus sp. in detail. Combined with the polyphasic classification method, we conducted an in-depth analysis and discussion of microalgae. We have rewritten the Results section to describe the research results in detail. On the other hand, we also explain the obtained results in detail and with detailed explanations of its importance.

Comment 3: The discussion needs to be strengthened in the light of the relevant literature available on the Desmodesmus sp.

Response 3: Thank you for your valuable advice, which will make the revision of the article structure more specific. We rewrote the discussion section and cited important references to discuss the findings in detail.

Comment 4: The paper needs to corrected throughout to develop the flow and findings of the paper in the sequential pattern.

Response 4: Thank you for the detailed review. We carefully modified the methods, results, discussion and conclusion sections. By adding STM、SEM experiments and SNP、InDel mutation site analysis, combining the method of polyphasic taxonomy, the content of the paper was enriched and the logic was smoother.

Comment 5: The figures in the article is not clear, it needs to be redrawn by increasing the font.

Response 5: It is really true as Reviewer suggested that figures in the article is not clear. Therefore, we have made appropriate adjustments to the image.

Comment 6: The paper requires major modification.

Response 6: Thank you for taking the time to review our manuscript, as you said, there are many problems with this article and your suggestions are crucial to the revision of the manuscript. Therefore, in the process of rewriting the article, we consider your comments to enrich the article. We would like to take this opportunity to thank you for all your time involved and this great opportunity for us to improve the manuscript. We hope you will find this revised version satisfactory.

Reply to Reviewer #2

We are very grateful to your comments for the manuscript. According with your advice, we tried our best to amend the relevant part and made some changes in the manuscript. These changes will not influence the content and framework of the paper. All of your questions were answered below. 

Comments: “The authors describe their findings on a Desmodesmus species from Tibetan Yamdrok Lake in Tibet, China via a series of methods including strain isolation, molecular identification and determination of culture conditions. Totally speaking, the authors should clearly emphasize the novelty of this finding in the manuscript. Besides, the following comments are also for your references to improve the quality.”

.1. ‘sp.’ should not in italic form.

2. The resolution of figures is too low to provide information.

3. The data should be carried out in triplicate, and error bars should be provided.

4. In the M&M section, the general process of experimental methods should be concise by referring the published paper.

5. The initial value of TN in Figure 3c should be 200-260 ppm. Please check.

6. There is almost no discussion within the manuscript. The authors should emphasize why they perform this work. What is the significance?

7. The format of the references needs to be unified.

8. There are spelling, grammar and formatting errors in this article, and it should be applied to a professional editing service for language improvement.

Comment 1: ‘sp.’ should not in italic form.

Response 1: Thank you for the detailed review. We have carefully and thoroughly proofread the manuscript to correct all the grammar and typos. We rewritten the article according to the review comments. Editage for its linguistic assistance during the preparation of this manuscript.

Comment 2: The resolution of figures is too low to provide information.

Response 2: Thank you for your valuable advice. It is really true as Reviewer suggested that figures in the article is not clear. Therefore, we have made appropriate adjustments to the image.

Comment 3: The data should be carried out in triplicate, and error bars should be provided.

Response 3: We are very sorry for our negligence of describe. In fact, we used 3 biological replicates, and added to the Methods section of the manuscript，and we have provided error bars in the figure.

Comment 4: In the M&M section, the general process of experimental methods should be concise by referring the published paper.

Response 4: It is really true as Reviewer suggested that referring the published paper in the article is missing. We have improved the M&M section.

Comment 5: The initial value of TN in Figure 3c should be 200-260 ppm. Please check.

Response 5: Thank you very much for your professional review. I'm very sorry that the initial total nitrogen value was not shown on the way during the mapping process. We made corrections. See the picture for details.

Comment 6: There is almost no discussion within the manuscript. The authors should emphasize why they perform this work. What is the significance?

Response 6: Thank you for the detailed review. We carefully modified the methods, results, discussion and conclusion sections. By adding STM、SEM experiments and SNP、InDel mutation site analysis, combining the method of polyphasic taxonomy, the content of the paper was enriched and the logic was smoother.

Comment 7: The format of the references needs to be unified.

Response 7: Thank you for your valuable advice, which will make the revision of the article structure more specific. We rewrote the references section according to the requirements of the journal.

Comment 8: There are spelling, grammar and formatting errors in this article, and it should be applied to a professional editing service for language improvement.

Response 8: Thank you for the detailed review. We have carefully and thoroughly proofread the manuscript to correct all the grammar and typos. We rewritten the article according to the review comments. Editage for its linguistic assistance during the preparation of this manuscript.

I wish this revision will be acceptable for publication in your journal.

Thank you for your consideration. I am looking forward to hearing from you.

Yours Sincerely,

Wang Jinhu

Address: Lhasa, China.

Email: phudor@vip.163.com

Tel: +8613618465558

---

## [Editor Report · Decision Letter 1]

26 Sep 2022

A new Desmodesmus sp. from the Tibetan Yamdrok Lake

PONE-D-22-12548R1

Dear Dr. Duo Bu,

We’re pleased to inform you that your manuscript has been judged scientifically suitable for publication and will be formally accepted for publication once it meets all outstanding technical requirements.

Kind regards,

Vandana Vinayak, PhD

Academic Editor

PLOS ONE